# Large non-thermal contribution to picosecond strain pulse generation using the photo-induced phase transition in VO$_2$

Iaroslav A. Mogunov [1]✉, Sergiy Lysenko [2], Anatolii E. Fedianin[1], Félix E. Fernández [2], Armando Rúa [2], Anthony J. Kent [3], Andrey V. Akimov [3] & Alexandra M. Kalashnikova [1]

Picosecond strain pulses are a versatile tool for investigation of mechanical properties of meso- and nano-scale objects with high temporal and spatial resolutions. Generation of such pulses is traditionally realized via ultrafast laser excitation of a light-to-strain transducer involving thermoelastic, deformation potential, or inverse piezoelectric effects. These approaches unavoidably lead to heat dissipation and a temperature rise, which can modify delicate specimens, like biological tissues, and ultimately destroy the transducer itself limiting the amplitude of generated picosecond strain. Here we propose a non-thermal mechanism for generating picosecond strain pulses via ultrafast photo-induced first-order phase transitions (PIPTs). We perform experiments on vanadium dioxide VO$_2$ films, which exhibit a first-order PIPT accompanied by a lattice change. We demonstrate that during femtosecond optical excitation of VO$_2$ the PIPT alone contributes to ultrafast expansion of this material as large as 0.45%, which is not accompanied by heat dissipation, and, for excitation density of 8 mJ cm$^{-2}$, exceeds the contribution from thermoelastic effect by a factor of five.

[1] Ioffe Institute, St. Petersburg, Russia 194021. [2] Department of Physics, University of Puerto Rico, Mayaguez, PR 00681, USA. [3] School of Physics and Astronomy, University of Nottingham, Nottingham NG7 2RD, UK. ✉email: mogunov@mail.ioffe.ru

Excitation of opaque materials with ultrashort laser pulses results in generation of ultrafast dynamical strain and emission of pico- and subpicosecond strain pulses[1]. This phenomenon has become an essential instrument for *picosecond ultrasonics* (for a review see, e.g., ref. [2]) allowing nanometer resolution in acoustic imaging and sensing mechanical properties of objects ranging from ordered solids[3] to liquids[4], polymers[5] and single cells[6,7]. Advantages provided by picosecond strain pulses are currently being actively explored for in-situ monitoring of chemical reactions[8], and ultrafast control of electronic[9], optical[10], and spintronic[11] devices.

The crucial element for picosecond ultrasonics is a light-to-strain transducer which transforms femtosecond optical excitation into a strain pulse. The most common materials for designing transducers are conventional metals[12–14] and semiconductors[1,15,16]. The amplitude, temporal, and spatial evolution of the generated strain pulses are governed by specific mechanisms responsible for the transformation of optical excitation to stress in the transducer. For instance, in Al, Cr, Au, and other metals[12–14] the main contribution to optically generated stress comes from a thermoelastic effect when optically heated electrons transfer their energy to the lattice at a time less than 1 ps after excitation, leading to thermal expansion[1]. However, the thermoelastic effect is unavoidably accompanied by undesirable heat dissipation in the transducer, and ultimately, for laser pulse excitation densities ~10–100 mJ cm$^{-2}$, the temperature rises up to ~$10^3$ K, resulting in the destruction of the thermoelastic transducer. Even for much lower laser pulse excitation densities the heat can modify delicate specimens, e.g., living cells. The contributions to the generated strain from the electron gas, deformation potential in semiconductors[1,17,18] and screening of electric field by photocarriers in piezoelectric materials[19,20] do not solve completely the problem of transducer heating because a significant part of the absorbed energy is still converted into heat after electron relaxation. Even upon excitation of a semiconducting transducer by photons near the edge of the bandgap, heating still occurs at high fluences because of the Auger process[18]. As a result, it is still a challenge to generate picosecond strain pulses with reduced heating in a transducer with amplitude higher than 1%[21–26]. Therefore, the full practical potential of picosecond ultrasonic applications cannot be realized unless mechanisms, and related materials, for photo-induced strain generation are found which outperform existing ones in both efficiency and tunability, and allow for significant reduction of transducer heating.

Our proposal is to develop transducers from a strongly correlated material which exhibits first-order coupled structural and electronic phase transitions at a critical temperature $T_c$, and to utilize the fact that such transitions are essentially nonthermal and can occur on a subpicosecond timescale under photoexcitation[27–31]. Since the first-order phase transition itself requires energy, a certain fraction of energy from optical excitation is spent on ultrafast photo-induced phase transition (PIPT) without lattice heating. Since structural phase transformations are accompanied by pronounced and often complex changes of crystalline symmetry and lattice constants, the PIPT might trigger almost instantaneous stresses yielding generation of picosecond strain with sufficient amplitude. An unexplored question is whether the stress emerging during PIPT in strongly correlated materials is capable of providing a nonthermal contribution to the ultrafast strain, comparable or even exceeding the contribution from thermoelastic mechanism.

The main goal of the present work is to elucidate the ultrafast strain dynamics accompanying PIPT in a strongly correlated material transducer, identify its origin, and demonstrate its contribution to strain pulse injected into a bulk substrate. Here we report on generation of picosecond strain pulses upon femtosecond photoexcitation of epitaxial layers of vanadium dioxide, which

exhibits electronic (insulator-to-metal) and structural (monoclinic-to-rutile) phase transitions at $T_c = 340$ K[32,33], and ultrafast PIPT when excited with femtosecond laser pulses with fluence above a threshold $W_T$ of several mJ cm$^{-2}$ (for review see ref. [34]). We use the nonlinearity of strain pulse propagation through a thick sapphire substrate to extract the absolute value of the strain generated in the VO$_2$ layer. When VO$_2$ is initially in an insulating phase, the PIPT is found to provide a contribution to the photo-generated strain which is as large as 0.45% while the estimated strain pulse generated by thermoelastic effect alone has five times smaller amplitude for the same laser fluence of 8 mJ cm$^{-2}$. Finally, total strain observed in our experiments reaches ~1.5% at $W \sim 12$ mJ cm$^{-2}$ and is comparable with maximal strain amplitudes reported so far[21–25]. Owing to a pronounced nonthermal contribution from PIPT, the transducer heating accompanying such a strain generation is significantly reduced. We also show the difference in strain pulse generation when VO$_2$ is initially in insulating ($T < T_c$) or in metallic state ($T > T_c$), and find that in metallic VO$_2$ there is no additional contribution to the generated strain except for thermoelastic effect and deformation potential. While these contributions also yield a ~1.5% strain amplitude at $W \sim 12$ mJ cm$^{-2}$, the accompanying heating is ~30% larger compared with the case in which the transducer is initially in its insulating state. Altogether, our findings clearly demonstrate the great potential of materials with first-order phase transitions for picosecond ultrasonics, since they allow generation of picosecond strain pulses with significantly reduced heat dissipation in the transducer.

## Results

### Evaluation of photo-generated picosecond strain in VO$_2$ film.

Our experiments were designed to extract unambiguously the amplitude, polarization, and temporal shape of the strain pulse emitted upon excitation of the VO$_2$ film by a femtosecond laser pulse. For this purpose, two structures were prepared (see Supplementary Note 1) consisting of either 100- or 35-nm-thick epitaxial layer of VO$_2$ grown on a 290-μm-thick r-cut Al$_2$O$_3$ single-crystal substrate, and a 30-nm layer of polycrystalline Cr deposited on the side of the substrate opposite to the VO$_2$ layer (Fig. 1a). Epitaxial VO$_2$ on the r-cut sapphire has its $\mathbf{a}_{M1}(\mathbf{c}_r)$-axis along the substrate normal, where the indices for the lattice axes denote low-temperature monoclinic (M1) and high-temperature rutile (r) phases. The choice of VO$_2$ on r-cut Al$_2$O$_3$ is motivated by well-defined twin-free orientation of such films and the large change of the lattice constant along the $\mathbf{a}_{M1}(\mathbf{c}_r)$ axis for the thermally driven transition. The latter is reported to be $-1\%$ for the bulk[35] and $-0.4\%$ for a 120-nm film[36].

The VO$_2$ layer is excited by a 170-fs laser pulse focused to a spot of a 25-μm diameter with fluence $W$ and a central photon energy of 1.2 eV, which is above the material bandgap of 0.6 eV[34]. VO$_2$ serves as *a light-to-strain transducer* in which strain of magnitude $\varepsilon_0$ is generated on a picosecond timescale. The thick Al$_2$O$_3$ substrate serves as *a nonlinear strain analyzer* enabling evaluation of the strain pulse amplitude injected into it from the VO$_2$ layer, and the Cr film serves as a conventional *photoelastic detector*. The Cr film's optical properties are altered by the strain pulse due to the photoelastic effect[14], and are monitored in the time domain by measuring the changes of intensity $\Delta R(t)$ of a reflected probe pulse (see "Methods" section). Because of the closely matching acoustic impedances of VO$_2$, sapphire, and Cr, there are no strong multiple reflections of the strain pulses at the interfaces within the structure, which could obscure interpretation of the experimental results (see "Methods" section).

To illustrate how the sapphire acts as the analyzer of the generated strain, in Fig. 1b we show the calculated temporal profile of a strain pulse with an amplitude close to $\varepsilon_0/2$, injected

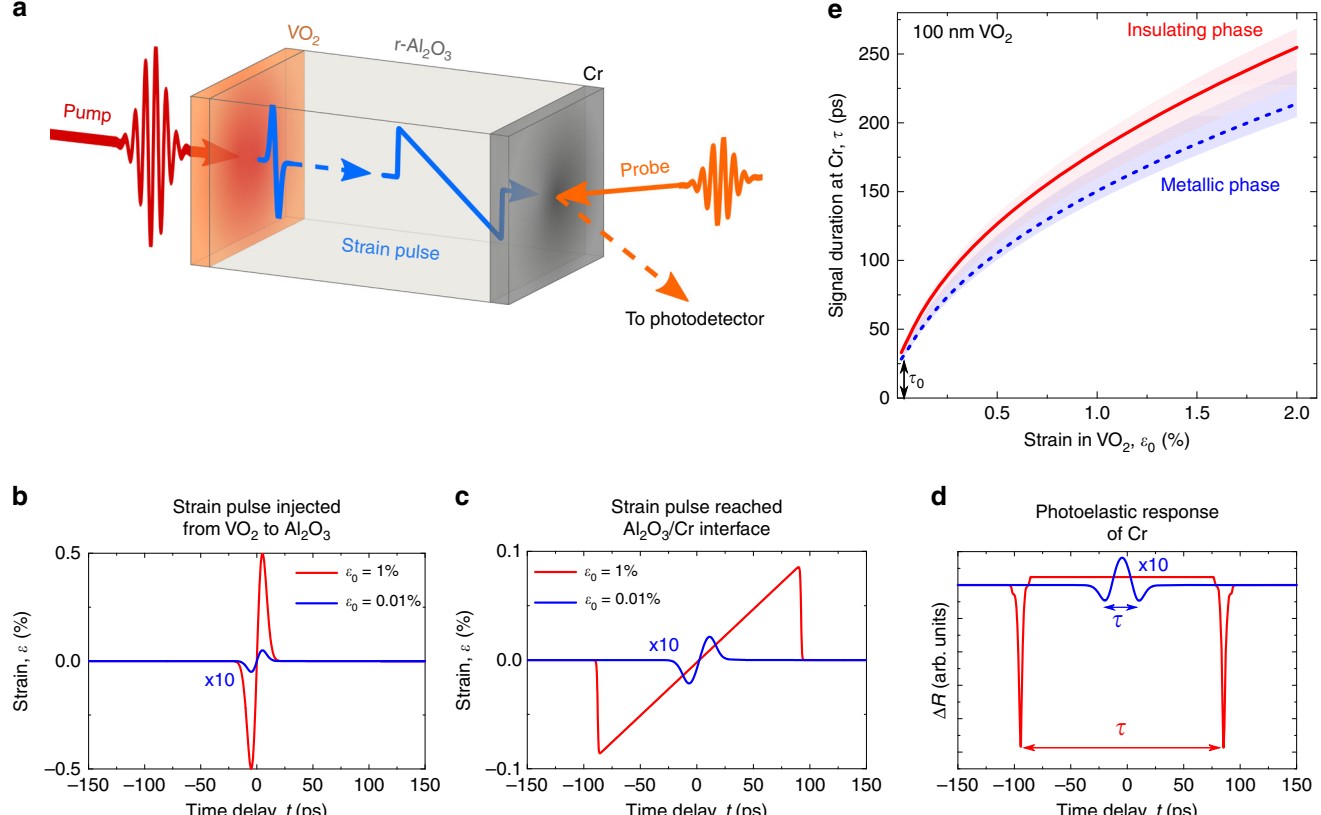

**Fig. 1 Principles of generation, propagation, and detection of the strain pulses generated upon photoexcitation in VO$_2$. a** Schematics of the experiment on detection of the strain pulses emitted into sapphire from the photoexcited VO$_2$ film. **b**, **c** Calculated temporal profiles of the strain pulses corresponding to the photo-generated strain in VO$_2$ of $\varepsilon_0 = 0.01\%$ (blue lines) and $\varepsilon_0 = 1\%$ (red lines), as initially injected into sapphire (**b**) and after propagation through the 290-µm-thick sapphire substrate (**c**). **d** Calculated photoelastic response of the Cr film induced by the two strain pulses shown in **c**. **e** Calibration curves $\tau(\varepsilon_0)$ providing the relation between photo-generated strain $\varepsilon_0$ in the insulating (red solid line) and metallic (blue dashed line) 100 nm VO$_2$ film and the duration $\tau$ of the photoelastic signal detected in the Cr film. Shaded areas in **e** indicate the uncertainly ranges found by varying sound velocities for VO$_2$ (see Supplementary Note 6 for details).

into Al$_2$O$_3$ from a 100 nm transducer whose acoustic properties mimic those of a $\mathbf{a}_{M1}(\mathbf{c_r})$-oriented VO$_2$. We assume that the strain pulse injection is a result of instantaneously photo-generated stress leading to a tensile strain in the transducer $\varepsilon_0 = 0.01\%$ (blue line) or $1\%$ (red line). In this case, the strain pulse injected from VO$_2$ into sapphire is bipolar consisting of a leading compressive part followed by a tensile part. Figure 1c, d show, respectively, the strain pulse transformed upon propagation through the 290-µm r-cut Al$_2$O$_3$ and the changes of the reflectivity $\Delta R(t)$ of Cr due to photoelastic effect. For the details of the simulations see the "Methods" Section. As can be seen, with an increase of the strain amplitude $\varepsilon_0$ the signal $\Delta R(t)$ acquires a shape containing two negative peaks with equal amplitudes, separated in time by the value $\tau$, which we subsequently refer to as the signal duration. For $\varepsilon_0 \to 0$ the signal duration $\tau$ approaches its minimum $\tau_0$ defined by the initial duration of the generated strain pulse, and by attenuation of high-frequency components of the pulse while propagating through sapphire and the Cr detector. Figure 1e shows the calculated dependencies of the signal duration $\tau$ on the value of $\varepsilon_0$ generated in the 100 nm VO$_2$ film being in insulating (red line) or metallic (blue line) phases. The relation between $\varepsilon_0$ and $\tau$ serves as a calibration for obtaining the strain $\varepsilon_0$ generated in VO$_2$ from the temporal characteristic of the measured photoelastic response of the Cr film. Analogous calibration curves $\tau(\varepsilon_0)$ are obtained for the case of the 35-nm-thick VO$_2$ sample.

**Picosecond strain pulses emitted from VO$_2$ above and below $T_c$.** Static and transient reflectivity studies (see Supplementary Notes 2 and 3) showed that the 100- and 35-nm-thick VO$_2$ layers undergo phase transitions at $T_c = 323$ K and $T_c = 315$ K, respectively. The PIPT takes place at excitation fluence $W$ between the threshold $W_T = 2$ mJ cm$^{-2}$ and saturation $W_S = 8$ mJ cm$^{-2}$ for the 100 nm film, and between $W_T = 0.7$ mJ cm$^{-2}$ and $W_S = 3.6$ mJ cm$^{-2}$ for the 35-nm film. In the main experiments, the femtosecond pump–probe setup was used for excitation of VO$_2$ and detection of the generated strain pulses in Cr, as described in the "Methods" section. The experiments were carried out at room temperature, $T = 295$ K, at which VO$_2$ films are initially in the insulating phase ($T < T_c$) and PIPT can be induced by optical excitation, and at $T = 350$ K ($T > T_c$), i.e., when VO$_2$ is in the metallic phase and thus no PIPT is excited. In both cases, the fluence of the pump pulses used to generate the strain in VO$_2$ was varied to cover the range including both $W_T$ and $W_S$.

Figure 2a, b shows the temporal traces of Cr film reflectance $\Delta R(t)$ measured for the sample with the 100-nm-thick VO$_2$ film at $T = 295$ K and $T = 350$ K, respectively. Time delay denoted as $t = 0$ in Fig. 2a, b corresponds to 28 ns after the moment the VO$_2$ film is excited by the laser pulse. This delay is equal to the time of propagation through sapphire substrate with longitudinal sound velocity. Therefore, the longitudinal strain pulse is detected. The main features of $\Delta R(t)$ are similar to those predicted in the simulations (Fig. 1d), i.e., two negative peaks in $\Delta R(t)$ with nearly

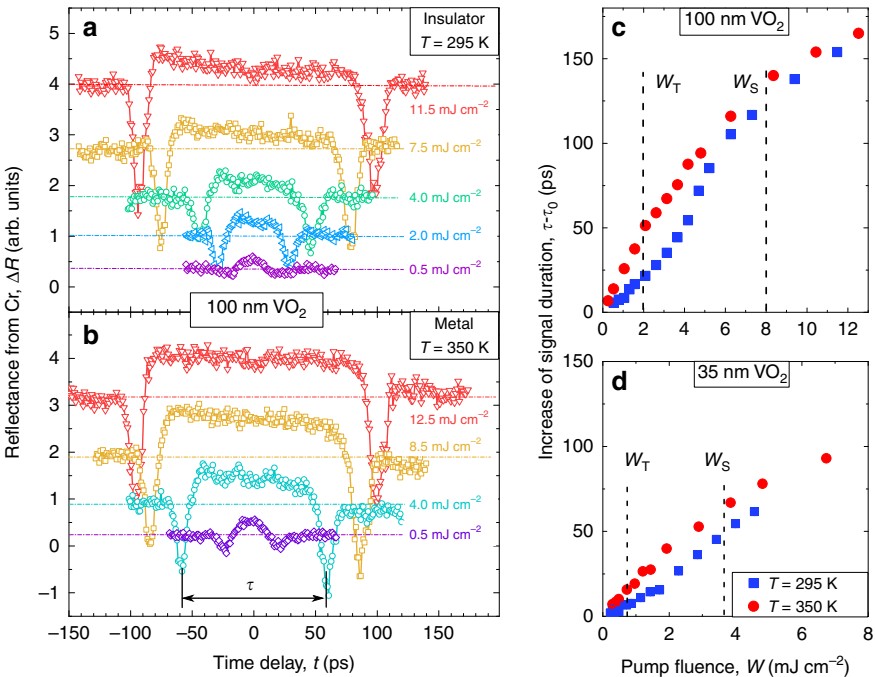

**Fig. 2 Experimentally detected strain pulses emitted upon photoexcitation of the VO₂ transducers. a, b** Transient reflectivity measured for the Cr film resulting from injection of the strain pulses generated by pulsed laser excitation, with fluence $W$, of the 100-nm-thick VO₂ transducer initially in the insulating (**a**) and metallic (**b**) phases. $t = 0$ corresponds to the time delay of 28 ns required for a strain pulse to propagate through the sapphire substrate. **c, d** The dependencies of signal duration $\Delta\tau = \tau - \tau_0$ on the fluence $W$ for the 100 nm (**c**) and 35 nm (**d**) thick VO₂ transducers, initially in insulating (blue symbols) and metallic (red symbols) phases. Vertical lines denote the PIPT threshold ($W_T$) and saturation ($W_S$) fluences, as obtained from the optical pump–probe experiments (Supplementary Note 3).

the same amplitudes are clearly seen. As the excitation density $W$ increases, the positive part of the signal between the peaks tends to form a plateau and the signal duration $\tau$ increases. This is a reliable evidence that the laser excitation results in tensile strain in VO₂. As a result, the longitudinal strain pulse emitted into sapphire has a bipolar shape such as that shown in Fig. 1b. We note that transducer contraction would yield strain pulses with reversed polarity and the duration of the signal detected in the Cr films would decrease with $W$[37]. In our experiments, for both initial sample temperatures, the signal duration $\tau$ between the peaks in $\Delta R(t)$ reaches the values of $\tau = 200 \pm 5$ ps exceeding those measured in sapphire earlier with metallic transducers[37]. Such high values of $\tau$ point at the high amplitude $\varepsilon_0 \sim 1.5\%$ of the tensile strain generated in VO₂, as can be readily seen from the calibration curve $\tau(\varepsilon_0)$ in Fig. 1e. As $W \to 0$ we obtain $\tau_0 \to 37$ ps at $T = 295$ K, and $\tau_0 \to 31$ ps at $T = 350$ K. In order to reveal if the PIPT provides any substantial contribution to the generated high amplitude strain, we examine in details how $\tau$ changes with increase of the excitation fluence in both insulating and metallic phases.

**Effect of PIPT on picosecond strain pulses emitted from VO₂.** Figure 2c shows the dependencies of the signal duration increase $\Delta\tau = \tau - \tau_0$ on the excitation fluence $W$ for both initial sample temperatures. Despite similar maximum values of $\Delta\tau$ reached at highest $W$, there is a striking difference between the fluence dependence of $\Delta\tau(W)$ for VO₂ excited while in its insulating or metallic phases. Thus, for the 100 nm VO₂ film excited in its insulating phase (blue symbols in Fig. 2c), $\Delta\tau(W)$ experiences a superlinear increase if the pump pulse fluence $W$ is in the range between PIPT excitation threshold, $W_T$, and saturation, $W_S$. No superlinear behavior is observed when the sample is initially in the metallic phase (red symbols in Fig. 2c). At $W > W_S$ the dependence of $\Delta\tau$ on $W$ found for the insulating phase of VO₂

shows a much smaller slope, which is similar to the one found for VO₂ in the metallic phase at similar fluences. Qualitatively similar behavior is found for the sample with the 35-nm-thick VO₂ transducer (Fig. 2d).

Observed differences in $\Delta\tau(W)$ for insulating and metallic VO₂ within the excitation fluence range $W_T$ to $W_S$ is the first evidence that the PIPT significantly contributes to strain generation. This is the main experimental result of the present work. To provide further insights into photo-induced strain upon PIPT, we analyze the transient strain $\varepsilon_0$ in VO₂ versus absorbed volume excitation energy density $J$. This analysis takes into account the change of the VO₂ optical properties in various initial phases (see Supplementary Note 4). The strain $\varepsilon_0(J)$ was calculated from the experimentally measured $\Delta\tau(W)$ data, using the calibration curves shown in Fig. 1e.

The dependencies $\varepsilon_0(J)$ for both films and both initial temperatures (i.e., samples at either insulating or metallic phase) are summarized in Fig. 3a, b. The way $\varepsilon_0$ increases with $J$ clearly depends on the initial state, insulating or metallic, of VO₂. As can be seen in Fig. 3a, the dependence $\varepsilon_0(J)$ is not linear when VO₂ is initially in the insulating phase. For absorbed energy density $J$ lower than the threshold value $J_T = 0.75 \times 10^8$ J m$^{-3}$, $\varepsilon_0(J)$ may be well approximated as a linear increase. However, when $J > J_T$, $\varepsilon_0(J)$ starts to increase more rapidly. As an excitation exceeds the saturation of PIPT, $J > J_S$, $\varepsilon_0(J)$ resumes linear dependence with a slope steeper than that for $J < J_T$. The $\varepsilon_0(J)$ dependence in the metallic phase is well described by a single linear growth (Fig. 3b) with a slope larger than for the insulating phase (Fig. 3a) at both $J < J_T$ and $J > J_S$. This confirms the preliminary conclusion about the evident role of PIPT in photo-generation of the strain made on the basis of the $\Delta\tau(W)$ dependence. There is a pronounced difference between absolute values of strain generated in the 100 nm (closed symbols) and 35 nm (open symbols) VO₂ transducers in both initial phases.

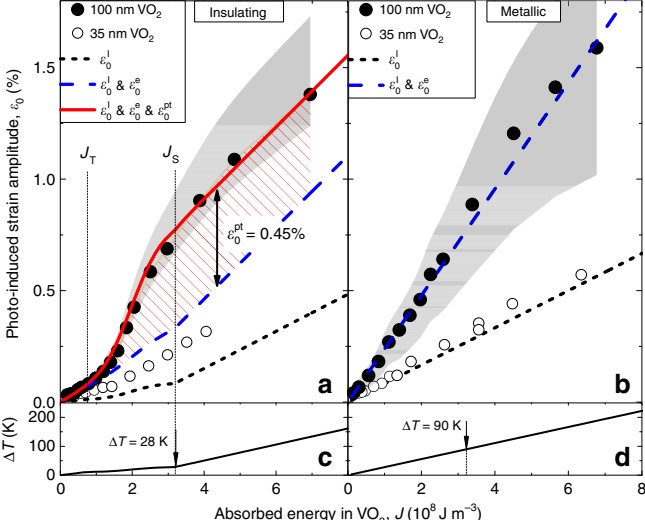

**Fig. 3 Strain generated in VO₂ transducers as a function of the energy deposited into transducers by laser pulses. a, b** Strain magnitude $\varepsilon_0$ versus the absorbed energy of the excitation laser pulses $J$, when the VO₂ film, initially in insulating (**a**) or metallic (**b**) phases, is excited. Data (symbols) are obtained from the experimental dependencies (Fig. 2c, d) and the calibration curves (Fig. 1e) for the 35 nm (open symbols) and 100 nm (closed symbols) VO₂ transducers. Gray shaded areas in **a**, **b** indicate the uncertainty ranges originating from the uncertainty in the calibration curves (Fig. 1e). The relation between the absorbed volume density $J$ and incident fluence $W$ is obtained from the reflection and transmission measurements performed under the corresponding experimental conditions (Supplementary Note 4). Lines show results of calculations of the photo-generated strain using Eq. (3), under the assumption of only thermoelastic contribution $\varepsilon_0^l$ (dotted black lines), thermoelastic and electronic contributions $\varepsilon_0^l$ and $\varepsilon_0^e$ (dashed blue lines), and all contributions (solid red line), including that from PIPT $\varepsilon_0^{pt}$ (dashed area in **a**). **c, d** Lower panels show calculated temperature increase in the 100 nm VO₂ film when excited in insulating (**c**) or metallic (**d**) phases.

## Discussion

We start the discussion with the case when VO₂ is in its insulating phase before optical excitation. In order to understand how PIPT affects the strain generation process, we consider three mechanisms which are expected to contribute to stress generation upon the optical excitation: thermoelastic effect; deformation potential; and a contribution from structural phase transition. We exclude the inverse piezoelectric effect since the crystal structure of VO₂ is centrosymmetric both below and above $T_c$. The contribution from structural phase transition is the main interest of the present work. As discussed above, the strain pulses injected to the sapphire substrate have a bipolar shape (Fig. 1b) and, thus, all contributions to the stress generated in VO₂ may be modeled by temporal step-like functions[1]. The net stress results in changes of the VO₂ film thickness $\Delta a$. This change happens during a time $\sim a/s_{VO_2}$, where $s_{VO_2} = 9740$ m s$^{-1}$ [38] is the longitudinal sound velocity in VO₂ in the direction perpendicular to the sample surface. For a 100-nm-thick transducer the time for the strain to emerge is then of ~10 ps. The corresponding strain amplitude is $\varepsilon_0 = \Delta a/a$.

The first contribution $\varepsilon_0^l$ to $\varepsilon_0$ is typical for conventional metals and semiconductors under high excitation densities and is due to the energy transfer from hot electrons to the lattice eventually resulting in heating by temperature $\Delta T$. This mechanism is classified as thermoelastic, and the related stress occurs at a time

delay less than 1 ps[1], resulting in a film expansion. The value of $\varepsilon_0^l$ can be calculated as:

$$\varepsilon_{0,i(m)}^l(J) = \alpha_{i(m)}\beta_{i(m)}J/C_{i(m)}, \qquad (1)$$

where $\alpha_{i(m)}$ is the linear thermal expansion coefficient in the direction perpendicular to the transducer plane, $\beta_{i(m)}$ is a fraction of total absorbed energy which is transferred to heat in a time ~1 ps, and $C_{i(m)}$ is the specific heat in the insulating (i) or metallic (m) phases. Related lattice heating is then $\Delta T = \beta_{i(m)}J/C_{i(m)}$. In the insulating phase, only energy exceeding the VO₂ bandgap (0.6 eV) is transferred to the lattice in a time less than 1 ps and we take $\beta_i = 0.42$. For the metallic phase, we assume that, similar to conventional metals, the total energy $J$ is active in ultrafast lattice "heating" and correspondingly $\beta_m = 1$.

The second contribution $\varepsilon_0^e$ to $\varepsilon_0$ comes from photoexcited electrons, and can be present in both phases. As a result of optical excitation of VO₂ in the insulating phase electrons with density $n_e$ occupy high-energy states having deformation potential $\Xi_i$, which results in the generation of stress[1,18,20]. For optical excitation of VO₂ being initially in the metallic phase the effect of electron gas heating is considered using the electron Grüneisen coefficient[17,20]. Taking into account that electron temperature rise is proportional to $J$, it is convenient to introduce a coefficient $\Xi_m$ which is the analog of deformation potential. Then the instantaneously generated stress results in a strain as follows:

$$\varepsilon_{0,i(m)}^e(J) = -\Xi_{i(m)}\frac{n_e(J)}{B} = -\Xi_{i(m)}\frac{\lambda}{Bhc}J, \qquad (2)$$

where $B$ is the bulk modulus, $\lambda$ is the excitation light wavelength, and $h$ and $c$ are Planck's constant and the speed of light, respectively. Both contributions described by Eqs. (1) and (2) are well known and widely discussed in the literature (for review see ref. [20]).

Now we discuss how the strain generation is affected by PIPT when laser pulses with excitation energy density $J > J_T$ excite VO₂ being initially in the insulating phase (Fig. 3a). First, between $J_T$ and $J_S$ the fraction of excited material which undergoes PIPT increases from 0 to 1, and can provide an extra contribution $\varepsilon_0^{pt}$ to the generated strain with a corresponding weight. At high excitation densities $J > J_S$ the contribution from PIPT to strain does not depend on $J$ and is equal to $\varepsilon_0^{pt}$. Second, it is essential that the PIPT, being a first-order phase transition, requires energy. Therefore, when calculating the thermoelastic contribution to the generated strain (Eq. 1), one has to take into consideration that the energy transfer to lattice "heating" is reduced by a value $\Delta J$ required for PIPT.

It is known from earlier studies that the PIPT threshold $J_T$ increases with the decrease of the initial sample temperature[34]. Furthermore, it is generally accepted that the metallic phase emerging as a result of PIPT can be stabilized at long time delays of hundreds of picoseconds if the laser energy deposited to the sample is sufficient to both heat the sample above $T_c$ and overcome the energy required for the transition. Then it is reasonable to assume that in our experiments at $J = J_S$ the temperature increase is $\Delta T = T_c - 295 = 28$ K. PIPT is nonthermal at the earlier stages (for a review on this issue see e.g., ref. [34]) and the metallic phase emerges at femtosecond timescale, which is faster than that required for the lattice "heating". Thus, we consider that the excess energy $J - \Delta J$ at the high excitation densities, $J \geq J_S$, yields the lattice "heating" and thermoelastic contribution $\varepsilon_{0,m}^l$ related to the photo-induced metallic phase only. Therefore, at $J = J_S$ the temperature increase can be found as $\Delta T = \beta_m(J_S - \Delta J)/C_m = 28$ K, yielding $\Delta J \sim 2.2 \times 10^8$ J m$^{-3}$ for the 100-nm-thick VO₂. This value is, in fact, close to the value of the latent heat $2.35 \times 10^8$ J m$^{-3}$ known for the phase transition in VO₂ at thermal equilibrium[39].

Following the arguments considered above, we write an equation which describes the dependence of $\varepsilon_0$ on $J$ including all contributions. The increase of the fraction of the excited material undergoing the PIPT from 0 to 1 as the absorbed energy increases from $J_T$ to $J_S$ is related to a distribution of nucleation sites in the sample versus energy which was approximated by the Gaussian error function centered at $J_0 = (J_T + J_S)/2$, with a dispersion parameter $\sigma_0$. Due to the various inhomogeneities in VO$_2$[40–42] we also take into account Gaussian distributions of $J_T$ and $J_S$ with narrower dispersions $\sigma_T$ and $\sigma_S$, respectively. Then the general expression for the strain generated upon excitation of the VO$_2$ in the insulating phase takes a form (see Supplementary Note 5 for details):

$$\varepsilon_0(J) = 0.5 \cdot \left[1 - \operatorname{erf}\frac{J - J_T}{\sqrt{2}\sigma_T}\right] \cdot \varepsilon_{0,i}^{l}(J)$$
$$+ 0.5 \cdot \left[1 - \operatorname{erf}\frac{J - J_T}{\sqrt{2}\sigma_T} \operatorname{erf}\frac{J - J_0}{\sqrt{2}\sigma_0}\right] \cdot \varepsilon_{0,i}^{l}(J_T)$$
$$+ 0.5 \cdot \left[1 + \operatorname{erf}\frac{J - J_S}{\sqrt{2}\sigma_S}\right] \cdot \varepsilon_{0,m}^{l}(J - \Delta J)$$
$$+ 0.5 \cdot \left[1 - \operatorname{erf}\frac{J - J_S}{\sqrt{2}\sigma_S} \operatorname{erf}\frac{J - J_0}{\sqrt{2}\sigma_0}\right] \cdot \varepsilon_{0,m}^{l}(J_S - \Delta J)$$
$$+ 0.5 \cdot \left[1 + \operatorname{erf}\frac{J - J_0}{\sqrt{2}\sigma_0}\right] \cdot \varepsilon_0^{pt} + \varepsilon_{0,i}^{e}(J). \quad (3)$$

In the calculations, we used the values $J_T = 0.75 \times 10^8$ J m$^{-3}$, $J_S = 3.16 \times 10^8$ J m$^{-3}$, $\sigma_0 = (J_S - J_T)/4$, and $\sigma_T = \sigma_S = (J_S - J_T)/40$. The Eq. (3) allows successful reproduction of all major features of the experimental dependence $\varepsilon_0(J)$ for the 100-nm-thick VO$_2$, as shown by the red solid line in Fig. 3a. The only free parameters in the calculations are the values of the deformation potentials, which are unknown for VO$_2$ in both phases, and the value of the PIPT-related strain $\varepsilon_0^{pt}$ (see "Methods" section). The black dotted line in Fig. 3a shows only the contribution from the thermoelastic effect. It is seen that the strain generated by thermoelastic effect when all VO$_2$ domains have undergone PIPT, $\varepsilon_0^{l}(J = J_S)$, is ~10 times smaller than values of strain $\varepsilon_0(J = J_S)$ obtained from the experimental data in the 100 nm VO$_2$ film. There is also a noticeable difference between measured $\varepsilon_0$ and $\varepsilon_0^{l}$ for small values of $J < J_T$ which points to a nonzero deformation potential mechanism for strain generation not related to PIPT. The blue dashed line shows the dependence $\varepsilon_0(J)$ which includes the contribution from electrons taking $\Xi_i = -4$ eV for all $J$ along with the thermoelastic one. It is seen in Fig. 3a that this blue line fits the experimental data well for low excitation densities, below PIPT threshold $J_T$. However, at $J > J_T$ the discrepancy between modeled blue curve and experimental data increases, and at $J = J_S$ the measured value of strain is twice the model value without contribution from PIPT. Ultimately, to get a good agreement in the whole range of $J$, we need to include the contribution $\varepsilon_0^{pt}$ from the PIPT. We obtain a good agreement with the experimental results taking $\varepsilon_0^{pt} = +0.45^{+0.19}_{-0.05}\%$ possessing the same sign as $\varepsilon_0^{l}$ and $\varepsilon_0^{e}$ (red solid line in Fig. 3a).

As can be seen, the PIPT contribution $\varepsilon_0^{pt}$ to the optically generated strain, marked as a dashed area on Fig. 3a, is important in the VO$_2$ transducer. The value of $\varepsilon_0^{pt}$ in the 100-nm-thick film at $J = J_S$ is roughly the same as the total from both thermal and electron contributions and is five times higher than the contribution from the thermoelastic effect.

We further compare the strain generated in the insulating VO$_2$ (Fig. 3a) with the one observed in the metallic phase (Fig. 3b). $\varepsilon_0(J)$ in this situation demonstrates a linear dependence, which

can be well described by combined thermoelastic and electronic contributions, with deformation potential set as $\Xi_m = -8$ eV.

Figure 3c, d show the calculated photo-induced increase of the transducer temperature for the 100 nm VO$_2$ film, illustrating an advantage of the PIPT-induced strain generation. When the VO$_2$ is excited being initially in its insulating phase (Fig. 3c), the temperature rise $dT/dJ$ decreases significantly between $J_T$ and $J_S$, while the generated strain still increases due to nonthermal PIPT-related $\varepsilon_0^{pt}$ and the electronic $\varepsilon_0^{e}(J)$ contributions. As a result, at $J = J_S$ photo-generated strain reaches ~0.8% in both insulating and in metallic VO$_2$ transducers, while the estimated temperature increase $\Delta T$ due to optical excitation is three times lower for insulating VO$_2$ than in its metallic phase.

For the 35 nm VO$_2$ film in both initial phases, the absolute values of the generated strain are considerably lower than those obtained in the thicker film (Fig. 3a, b). Furthermore, the features between $J_T$ and $J_S$ for the thinner film are much less pronounced, indicating that the PIPT contribution $\varepsilon_0^{pt}$ is considerably lower in this case. The suppression of the PIPT contribution to the photo-generated strain in the thinner VO$_2$ film can be ascribed to large static stresses present in this case due to the lattice mismatch between VO$_2$ and r-cut Al$_2$O$_3$[43], which also causes the significantly reduced transition temperature (see Supplementary Note 2). The mismatch is most important in VO$_2$ atomic layers located close to the VO$_2$/sapphire interface, and the resulting stress is relaxed due to misfit dislocations only for films thicker than ~80 nm[44], for which lattice and elastic parameters become closer to those of the bulk material.

Therefore, the suggested model for the photo-generated strain $\varepsilon_0$ in VO$_2$ allows us to identify the contribution $\varepsilon_0^{pt}$ originating from PIPT, which appears to be very pronounced in the 100-nm-thick VO$_2$ and somewhat suppressed in the thinner, 35 nm, film with high misfit strain. This contribution adds to the conventional thermoelastic and deformation potential contributions and is present only if VO$_2$ is excited in its insulating phase with the optical pump at a fluence exceeding the PIPT threshold.

The experiments show that the contribution of PIPT to the strain pulse is positive, i.e., $\varepsilon_0^{pt} > 0$, which means that the film expands during PIPT, at least over the time of the strain pulse generation $a/s_{VO_2} \sim 10$ ps. This result is different from the behavior in thermal equilibrium, when the lattice constant (in the direction of the rutile axis) is smaller ($\varepsilon_0^{pt} < 0$) in the metallic than in the insulating phase by 1% in bulk[35] or 0.4% in a 120 nm film[36] and, correspondingly, our experiments suggest that during PIPT the lattice reaches the final compressed state non-monotonically. The fast initial expansion of VO$_2$ thin film along $\mathbf{a}_{M1}$ axis up to 0.4% due to PIPT was also observed previously by ultrafast X-Ray diffraction[45]. Therefore, we conclude that after PIPT the VO$_2$ film expands along the rutile axis during a short time $\sim a/s$ and then slowly shrinks. Such non-monotonic behavior has been observed earlier in some electron diffraction experiments[46–51].

In conclusion, we have studied the generation of picosecond strain pulses in VO$_2$ photoelastic transducers. There is a large contribution to the generated strain pulse from the VO$_2$ lattice reconstruction during ultrafast PIPT when VO$_2$ is optically excited while being initially in its insulating phase. This contribution has a value of ~0.45% in a 100-nm-thick VO$_2$ transducer and is not present when the same film is excited while in its metallic phase, i.e., at elevated temperature $T > T_c$. The crucial result is that the contribution from PIPT is not accompanied by temperature rise. The net effect of PIPT and deformation potential on optically generated strain exceeds the thermoelastic contribution by an order of magnitude and allows the generation of strain amplitude ~0.8% for excitation density ~8 mJ cm$^{-2}$ with a temperature rise as small as 28 K.

Designing $VO_2$ nanostructures with sharp PIPT, i.e., with threshold and saturation fluences close to each other, $W_T \approx W_S$[52], would allow strain generation with negligible lattice heating when working at $T \approx T_c$[49] since the energy in this case is fully spent for the PIPT excitation. It would further enable fine tuning of the generated strain pulses' parameters by switching PIPT on or off using various means, such as varying excitation fluence in a narrow range, applying voltage[53], or strain[42,54–56]. Using transducers grown on differently oriented sapphire or other substrates may enable control over the direction in which the largest nonthermal strain generation occurs thus opening a pathway for a further optimization. This and the complex path which the lattice takes following photoexcitation[46,50] suggests that $VO_2$ transducers with different orientations grown on various substrates may allow the generation of, not only compressive-tensile strain pulses, but also shear ones. Furthermore, the progress in growth of high-quality $VO_2$ films[57,58] enables tuning their transition temperature and even stabilizing various phases at ambient conditions, which can be further utilized for PIPT-induced generation of strain pulses with various parameters. Implementation of $VO_2$ transducers grown on different substrates providing optimal conditions for nonthermal strain generation into modern picosecond acoustic studies of nano-objects is feasible since there are no strict requirements regarding coupling of the latter to a particular substrate.

## Methods

**Picosecond acoustic experiment**. The experiments are performed using the conventional scheme for a picosecond acoustics experiment (see Fig. 1a): The laser source is a Yb:KGd(WO$_4$)$_2$ regenerative amplifier with a pulse duration of 170 fs, central wavelength $\lambda = 1028$ nm and a repetition rate of 5 kHz. The optical pump pulse with a fluence $W$ is focused into a 110-μm spot on a $VO_2$ transducer. The value of $J$ is obtained from $W$ taking into account the measured reflection and transmission of the pump beam (Supplementary Note 4). The resulting strain pulse reaches the Cr film deposited on the opposite side of the substrate in 28 ns. The probe optical pulse is delayed with respect to the optical pump pulse by $(t + 28$ ns) and is focused at the 30 nm Cr film into a spot with diameter 25 μm. The time delay $t$ is controlled by an optomechanical delay line. The pump intensity is modulated at a frequency of 625 Hz with a mechanical chopper synchronized with the laser source. In experiments we monitor the intensity of the probe beam reflected from Cr. The sample is mounted on a copper plate with a heater and thermocouple allowing control of the temperature in the range $T_0 = 295–400$ K.

**Simulations of propagation and detection of the strain pulse**. We consider the temporal profile of the stress generated in the $VO_2$ film upon optical excitation as a step function, which is typical for the thermoelastic mechanism in metals[1]. For the $VO_2$ films of thicknesses $a = 100$ nm or 35 nm used in the experiments, it is reasonable to assume homogeneous distribution of the generated stress along their depth. Generated stress yields the film expansion $\Delta a$ along the film's normal on a timescale of $\sim a/s_{VO_2} \sim 10$ ps. The sound velocities $s_{VO_2}$ in the insulating and metallic phases of $VO_2$ along the $\mathbf{a}_{M1}$ and $\mathbf{c}_r$ axes are 9740 and 9480 m s$^{-1}$, respectively, as calculated from the elastic constants and mass density[38]. The strain $\varepsilon_0 = \Delta a/a$ generated in the $VO_2$ results in a bipolar strain pulse $\varepsilon(t, x = 0)$ injected into the sapphire substrate[1]. The amplitude of this injected strain pulse is $\sim\varepsilon_0/2$ because of the closely matched acoustic impedances $z$ of $VO_2$ and sapphire $z_{VO_2}/z_{Al_2O_3} = 1.03$. The latter also results in a simple shape of the injected pulse, with no ringing.

We performed calculations for various strain magnitudes $\varepsilon_0 = \Delta a/a$ generated in the $VO_2$ transducer, corresponding to different pump pulse fluences incident on the transducer. We approximate the shape of the strain pulse injected into the sapphire $\varepsilon(t, x = 0)$ as the derivative of a Gaussian function[2], as shown in Fig. 1b for the 100 nm film.

Propagation of the strain pulse through r-cut sapphire with thickness $d = 290$ μm is described by the Korteweg de Vries—Burgers (KdVB) equation

$$\frac{\partial \varepsilon(t,x)}{\partial t} = -\frac{\kappa}{2\rho s_{Al_2O_3}} \varepsilon(t,x) \frac{\partial \varepsilon(t,x)}{\partial y} - \beta \frac{\partial^3 \varepsilon(t,x)}{\partial y^3} + \frac{\eta}{2\rho} \frac{\partial^2 \varepsilon(t,x)}{\partial y^2}, \quad (4)$$

where $y = x - s_{Al_2O_3} \cdot t$, longitudinal sound velocity $s_{Al_2O_3} = 10800$ m s$^{-1}$, nonlinear parameter $\kappa = -3.51 \times 10^{12}$ N m$^{-2}$ was calculated using elastic constants of $Al_2O_3$[59], the acoustic phonon dispersion parameter $\beta = 3.5 \times 10^{-17}$ m$^3$ s$^{-1}$ and the viscosity $\eta = 6 \times 10^{-4}$ N s m$^{-2}$ were taken to be those for c-$Al_2O_3$[60], and $\rho = 3980$ kg m$^{-3}$ is the mass density of sapphire. The KdVB equation was solved numerically in time and space using the finite-difference method.

After propagation through the sapphire substrate, the strain pulse is injected into the 30-nm Cr film. The acoustic impedance of Cr is almost equal to that of sapphire, which suppresses reflections at the interface. The Cr film acts as a photoelastic detector of the strain $\varepsilon$, which alters its dielectric permittivity as

$$\Delta \epsilon = -n^4 p \varepsilon, \quad (5)$$

where $n$ is the refractive index of the medium, $\epsilon$ and $p$ are dielectric permittivity and photoelastic constant, respectively. The changes of the reflectivity $\Delta R(t)$ of Cr upon propagation of the strain through it were calculated using transfer matrices and Green function formalism. Using the steps described above we have calculated the dependencies $\tau(\varepsilon_0)$ (Fig. 1e).

**Calculations of the contributions to the photo-generated strain**. The parameters for the insulating (i) and metallic (m) phases of $VO_2$ are: $\alpha_i = 1 \times 10^{-5}$ K$^{-1}$; $\alpha_m = 3 \times 10^{-5}$ K$^{-1}$;[35] $C_i = 3 \times 10^6$ J K$^{-1}$; $C_m = 3.6 \times 10^6$ J K$^{-1}$ [61]. Analogously, the purely electronic contribution $\varepsilon_0^l(J)$ has been calculated using $B = 4.3 \times 10^{11}$ Pa[38] taken to be the same in insulating and metallic phases. The deformation potential $\Xi$ was chosen in such a way that thermoelastic and electronic contributions describe well the experimental dependence $\varepsilon_0(J)$ at $J < J_T$ for the 100-nm-thick $VO_2$ film when the latter is in the insulating phase (Fig. 3a), and in the whole range of $J$ when the $VO_2$ is initially in the metallic phase (Fig. 3b).

## Data availability

Raw pump-probe data presented on Fig. 2a, b as well as processed data (Fig. 3a, b) and Supplementary data that support the findings of this study are available in Mendeley Data with the identifier https://doi.org/10.17632/z3g5f2fng9.1 (ref. [62]).

## Code availability

Custom MATLAB codes used for calculations in the study are available from the corresponding author upon reasonable request.

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

## Acknowledgements

We thank V.S. Levitskiy for the ellipsometry measurements and V.I. Kozub for enlightening discussions. We thank D. Schemionek and A.V. Scherbakov for help with sample preparation. This work was supported by the collaborative grant of Russian Foundation for Basic Research (grant No. 17-52-10015) and the Royal Society (grant No. IEC\R2\170217), as well as by the Engineering and Physical Sciences Research Council (grant No. EP/M016161/1). S.L., F.F., and A.R. were supported in part by the US Army Research Laboratory and the US Army Research Office under contract No. W911NF-15-1-0448. Calculations were partly supported by the RFBR (grant No. 19-52-12063).

## Author contributions

A.M.K. and A.V.A. suggested the idea. S.L., F.F., and A.R. grew and characterized the samples. Ia.A.M., A.J.K., and A.V.A. performed experiments. Ia.A.M., A.M.K., and A.V.A. analyzed the experimental data, A.V.A., A.E.F., and Ia.A.M. performed modeling. All the authors discussed the results and contributed to writing the paper.

## Competing interests

The authors declare no competing interests.
