## [Peer Review File · Nature Communications]

Reviewers' comments:

Reviewer #1 (Remarks to the Author):

This is an interesting article on a novel approach for generation of large amplitude picosecond strain pulses using the photo-induced phase transition in vanadium oxide. The article is well written and well organized. However I am not fully convinced by the main argument put forth in this manuscript. The authors contend that using VO₂ as a transducer film for generation of large amplitude picosecond strain pulses will circumvent issues associated with thermally induced deterioration of metallic and semiconducting transducer films. I would certainly say this is the case for high quality, single crystal films used in this work. However, the quality of the film is highly dependent on the substrate. For this approach to have broad appeal, it should be applicable to a wide range of substrates. It is expected that growing high quality films on arbitrary substrates will be a challenge and in some cases not possible. It is known that the transition temperature for VO₂ broadens significantly and the change in resistivity decreases markedly for low quality films. In turn, I would expect that the quality of the film will influence the magnitude of the strain generated via the PIPT effect. The article would be strengthened by a short discussion of how this approach might be applied to arbitrary substrates. The authors should also address the following issues:

Title1 – The thermoelastic contribution to the signal is 20% so using “non-thermal is misleading.

Title2 – This article is really about generating large amplitude picosecond strain pulses. This should be reflected in the title.

Abstract1 – Destruction of metallic transducer layers not an issue for small amplitude picosecond strain pulses. For instance, imaging using time domain Brillouin scattering is routinely performed without causing any damage to the metallic transducer film. This really only becomes an issue for generation of large amplitude strain pulses.

Abstract2 – The authors neglect mentioning the piezoelectric generation mechanism.

Abstract and into – More emphasis should be placed on generation of large amplitude strain pulses.

Page 2 (end of second paragraph) – The authors mention that transducer heating is still an issue with semiconducting transducer films. This is not the case if one can pump to the edge of the conduction band. Heat is generated by decay of photoexcited electrons within the conduction band.

Page 4 (middle of first paragraph) – The spot size should be given here. Also it should be mentioned that above bandgap photons are used for excitation.

Page 5 (second paragraph) – Information regarding the pump probe setup (laser type, wavelength, and rep rate) should be placed in the methods section.

Reviewer #2 (Remarks to the Author):

In this manuscript, the authors report the photo-induced phase transition (PIPT)PIPT induced strain in a VO₂ thin film and its relations with thermoelastic and electronic induced strain. In the experiment, femtosecond optical pulses are used to generate pico-second strain pulses in VO₂, the strain pulses propagate, disperse in Al₂O₃ film, and are optically detected using a Cr film. The so-called signal duration between two dips in optical reflectivity is proportional to the strain. It allows

the measurement of strain generated in VO₂ which is as high as 1.5% according to simulation results. By comparing the difference between strain pulse generations in insulating and metallic VO₂ samples, the authors highlighted that it's PIPT instead of a thermal effect that dominates the strain pulse generation mechanism, which can result in 0.45% of strain.

The idea of strain generation via PIPT is not very new. For example, "Photoinduced Strain Release and Phase Transition Dynamics of Solid-Supported Ultrathin Vanadium Dioxide" (<https://www.nature.com/articles/s41598-017-10217-0.pdf>) has discussed this process. There is no significant new physics in the PIPT induced strain, which is an expected result. However, whether phase-change materials can be practically used as an alternative of conventional metallic transducer is not well known, partly due to the limited quantitative analysis of this process. In addition, it is generally preferable to have a "clean dynamical strain-induced phenomena" rather than thermally induced strain. A quantitative understanding of PIPT generated strain is needed for the application of PIPT as a mechanism to generate strain pulse.

The value of this work is to quantify and isolate the strain contribution from multiple mechanisms. The author did a great job to normalize the strain to absorbed energy (Fig. 3) which clarifies the involvement of PIPT during the process, however, it is disappointing that PIPT seems not practically generate higher strain than that in photoexcited metallic VO₂. This can be seen in fig.2 c,d. The red curve (without PIPT) generates higher (when the pump fluence is in the WT-WS range) or similar (when the fluence is larger than WS) strain than the blue curve (with PIPT). After considering the calibration curve in Fig. 1e, it is clear the involvement of PIPT does not produce larger strain than that generated in metallic VO₂. This finding significantly limits the impact of the use of PIPT for strain generation. Secondly, I am concern about the accuracy of the estimated strain value since the strain detection via optical reflectivity is indirect (See comments 3 below). Therefore, I am afraid I won't be able to recommend this work to be published in Nature Communication due to its limited impact.

Nevertheless, this work reports interesting experiments, high-quality data, and the manuscript is well written. The methodology laid out in this work can be valuable to discuss similar process. I have a few comments that may help the authors to further improve the manuscript. I would suggest the authors consider submitting to Scientific Report. I would think the quality of this work has met the standard of Scientific Report.

1. In Eq. 1 of this work, the authors seem to imply that the "thermal expansion" induced strain occurs within 1 ps, which is not accurate. In the Thomsen paper (Ref.[1]), the stress is generated via photoexcitation, (Eq. 4 in Ref. [1]). The stress can be created within 1 ps but it takes time to build the strain, which is usually limited by the sound velocity. Although this process is somewhat discussed in the Method section, it is not clear how photoexcitation leads to strain. To describe the process appropriately, it is necessary that the authors start with photoinduced stress, then discuss strain.

2. The expansion of a axis as speculated in this work is not consistent with the equilibrium heating effect. The author has tried hard to argue this is plausible, but with limited experimental support. Why do the authors have to assign lattice expansion for the observed strain pulse? I would think lattice contraction can also produce strain.

3. It is not clear how accurate the estimated strain is based on optical reflectivity from Cr. The reflectivity only can infer the magnitude of the strain with a few assumptions. It is not a direct probe of absolute strain using x-ray or electron diffraction. I am not asking for time-resolved x-ray diffraction measurements but would like to know the error bar of this estimation. For example, how does the calibration curve in Fig.1e generated? Is that from simulation or experiments? Can the authors estimate the error bar of the claimed 0.45% strain arising from PIPT?

4. How about PIPT induced strain in VO₂ film growing on c-cut Al₂O₃? Does the crystal orientation

matter in this process? Some discussions on this subject might be helpful.

Reviewer #3 (Remarks to the Author):

This manuscript by Mogunov et al. analyzes the acoustic response of a thin film of VO₂ / Al₂O₃ following photoexcitation from both the insulating and the metallic phase. The acoustic dynamics, in particular the amount of out-of-plane strain created after photoexcitation, are measured by detecting strain induced changes in reflectivity in a Cr film deposited on the back side of the r-cut Al₂O₃ substrate. The authors observe significant photoinduced strain effects in the VO₂ film, on the order of 1.5%. In the case of photoexcitation from the insulating phase a significant fraction of the deposited energy density is taken up by the latent heat across the insulator-to-metal transition and does not contribute to lattice heating. The main point of the paper is that materials exhibiting first order structural transitions, such as VO₂, can be exploited as efficient light-to-strain transducers with significantly less risk of laser induced damage, since photoexcitation of these systems can lead to large amounts of strain in the material with moderate sample heating.

This manuscript presents a proof of concept for a very nice new idea, which expands the capabilities of photoinduced strain generation. The experiments are well designed and the results are well presented and analyzed. Nothing new is learned about the physics of VO₂ or of its phase transitions but that is not the point of the work, and the choice of this material to demonstrate the idea is quite appropriate. I believe this paper will be suitable for publication in Nature Communications after the few questions / comments below have been addressed.

Main comments:

- 1) On page 10 the authors state that "the metallic phase emerges at subpicosecond time scale, which is faster than that required for the lattice "heating"", so that for $J > J_s$ the "thermoelastic contribution ϵ_{0m}^I related to the photo-induced metallic phase only" needs to be considered. While there might be some debate about this, I think it's a reasonable simplification. However, shouldn't then $\Delta T = \beta m (J_s - \Delta J) / C_m = 28 \text{ K}$ be calculated using the insulating phase constants, instead of the metallic phase ones?
- 2) "This result is different from the behavior in thermal equilibrium, when the lattice constant (in the direction of the rutile axis) is 1 % smaller in metallic than in insulating phase" (p. 12): is this also true for the c_r axis of 100 nm thin VO₂ films deposited on r-cut Al₂O₃? The reference cited is for bulk samples (and those are also the references used in section 1 of the supporting information), and strained films could behave very differently. This should be verified (either in the literature or by measuring the static lattice constant variation with T for films under study) for the comparison to the ultrafast measurements to make sense.
- 3) Assuming there is indeed a discrepancy between the sign of ϵ^{pt} in static vs. dynamic measurements, what could be the reason for that? It is not clearly addressed by the authors. Here it might be helpful to refer to two papers by Gray et al. (PRL 116, 116403 (2016) and PRB 98, 045104 (2018)).

Minor points:

- i) In Fig. 2, $t = 0$ is not the same for all the time axes. This should be clarified for clarity, as it is it seems like the strain pulse on the Cr film arrives before the light pulse hits the VO₂. A 28 ns time offset (in the measured traces) is mentioned only in the methods.
- ii) Page 10: "It is natural to approximate the increase of the fraction" => worth providing a bit more detail on why this is "natural"? Perhaps relating it to the Avrami model (with exponent 2), which is very similar to erfc ?
- iii) "At an excitation close to the saturation of PIPT, $J \sim J_s$, $\epsilon_0(J)$ flattens off" (p. 7); "temperature rise stops between J_t and J_s " (p. 11): none of these is very precise, consider rephrasing.

iv) "the film expands during PIPT, at least over the first 10 ps" (p. 12): where does the 10 ps timescale come from?

v) "At first, after PIPT, the VO₂ film expands along the rutile axis during a short time $\sim a/s$ and then shrinks more slowly until thermodynamic equilibrium is reached." (p. 12): the shrinking part of the dynamics is not discussed before, how can it be seen from the data?

Reply to the Reviewer #1:

We thank the reviewer for a thorough reading of the manuscript and valuable comments. We appreciate that the reviewer finds our manuscript interesting, well written and well organized. However, although the reviewer correctly points out that we report on the novel approach for generating picosecond strain pulses, he/she suggests that the main emphasis in the title, abstract and introduction should be on the large strain amplitude demonstrated here.

In response to this comment of the reviewer we would like to stress that our main aim was actually to demonstrate a role of PIPT in generation of strain pulses. The large amplitude of the generated strain is, indeed, a remarkable result. However, we would like to draw attention of a readership to the PIPT-related mechanism of strain generation, which is novel and, most importantly, non-thermal. From this comment of the reviewer and a few comments of other reviewers we conclude that this message was not fully conveyed by the original manuscript. Therefore, we made the revisions according to all the reviewers' comments and suggestions which helped us to emphasize this message.

Reviewer comment: The article would be strengthened by a short discussion of how this approach might be applied to arbitrary substrates.

We thank the reviewer for this useful suggestion. Although the growth of high-quality VO₂ films is a challenging issue, remarkable progress has been made in this area. On the one hand, modern picosecond acoustics studies objects which are not strongly related to the substrate, e.g. biological cells, polymers, organic materials, colloids, graphene, van der Waals materials, etc. These specimens may be deposited on different substrates and sapphire is one of the most common for this. Thus, using VO₂ on sapphire as a transducer may be applied to many objects for the studies in picosecond ultrasonics. On the other hand, the effect of a substrate on a phase transition may be seen as an advantage, since the substrate related strain helps to tune the transition temperature even down to room temperature.

We have revised the conclusions section by adding the following: "This and the complex path which the lattice takes following photo-excitation [40, 49: X. He et al., *Sci. Rep.* **7**, 10045 (2017)] suggests that VO₂ transducers with different orientations grown on various substrates may allow the generation of, not only compressive-tensile strain pulses, but also shear ones. Furthermore, the progress in growth of high-quality VO₂ films [56: Yo. Higuchi et al., *Appl. Phys. Exp.* **11**, 085503 (2018), 57: K. Okimura et al., *J. Appl. Phys.* **111**, 073514 (2012)] enables tuning its transition temperature and even stabilizing various phases at ambient conditions, which can be further utilized for PIPT-induced generation of strain pulses with various parameters. Implementation of VO₂ transducers grown on different substrates providing optimal conditions for non-thermal strain generation into modern picosecond acoustic studies of nanoobjects is feasible since there are no strict requirements regarding coupling of the latter to a particular substrate." (#1, #30 in the List of changes).

Reviewer comment: Title1 – The thermoelastic contribution to the signal is 20% so using “non-thermal” is misleading.

We agree that thermoelastic contribution is still present in our experiment. However, this contribution does not dominate the signal, which we tried to reflect in the title. In order to address this fair comment of the reviewer, we revised the title as follows: "Large non-thermal contribution to generating picosecond strain pulses during ultrafast photo-induced phase transition in vanadium dioxide" (#2 in the List of changes).

Reviewer comment: Title2 – This article is really about generating large amplitude picosecond strain pulses. This should be reflected in the title.

We have deliberately omitted mentioning high overall strain amplitude in the title in order to highlight that the main physical result reported in the manuscript is the demonstration of significant non-thermal contribution to the strain pulse generation. Therefore, we find it more appropriate to include the phrase “Large non-thermal contribution” instead of “Large strain amplitude” into the title. (#2 in the List of changes).

Reviewer comment: Abstract1 – Destruction of metallic transducer layers not an issue for small amplitude picosecond strain pulses. For instance, imaging using time domain Brillouin scattering is routinely performed without causing any damage to the metallic transducer film. This really only becomes an issue for generation of large amplitude strain pulses.

We agree with the referee. However, even small heating on ~10 degrees may modify the object under investigation in acoustical micro- and nanoscopy, e.g. a living cell. We have revised the abstract by adding: “These approaches unavoidably lead to heat dissipation and a temperature rise which ultimately results in the destruction of the transducer when large strain amplitudes are required. Even at lower heating levels, delicate specimens, like biological tissues, may be modified.” (#3 in the List of changes).

Reviewer comment: Abstract2 – The authors neglect mentioning the piezoelectric generation mechanism.

We agree with the reviewer that the piezoelectric mechanism plays an important role in strain generation especially if one aims at obtaining shear strain pulses; and we did cite the relevant paper in the introduction [26: M. Lejman et al., *Nat. Commun.* **5**, 4301 (2014)] (#4, #30 in the List of changes). In order to address the reviewer’s comment we have revised the abstract as follows: “This excitation results in the generation of instantaneous stress due to thermoelastic effect, deformation potential, or inverse piezoelectric effect” (#5 in the List of changes). We also added to the introduction a reference to the paper where the acoustics of piezoelectric materials is discussed in details: “The contributions to the generated strain from the electron gas, deformation potential in semiconductors [1, 17, 18] and screening of electric field by photocarriers in piezoelectric materials [19: P. Babilotte, et al., *Appl. Phys. Lett.* **97**, 174103 (2010), 20] do not solve completely the problem of transducer heating because a significant part of the absorbed energy is still converted into heat after electron relaxation.” (#6, #30 in the List of changes). Further, we added the note to the Discussion section: “We exclude the inverse piezoelectric effect since the crystal structure of VO₂ is centrosymmetric both below and above T_c ” (#7 in the List of changes).

Reviewer comment: Abstract and into – More emphasis should be placed on generation of large amplitude strain pulses.

Indeed, the strain amplitude measured in our work is more than 1 % which is high. However, such high amplitude has been obtained in earlier works (see e.g. reference [21-24, 25: V. V. Temnov et al, *Nat. Commun.* **4**, 1468 (2013), 26]. Thus this technical result is mentioned, but not as the main result in the present manuscript. In the present manuscript we have put more emphasis on using PIPT to reduce heating when strain is generated. We added the reference [25] to the introduction (#30 in the List of changes).

Reviewer comment: Page 2 (end of second paragraph) – The authors mention that transducer heating is still an issue with semiconducting transducer films. This is not the case if one can pump to the edge

of the conduction band. Heat is generated by decay of photoexcited electrons within the conduction band.

We agree with this comment of the Reviewer. However, even upon excitation by the photons near the edge of the bandgap of a semiconducting transducer, the heating occurs at high fluences because of the Auger process [18]. We also note, that there are only few semiconductor structures which have the optical efficiency close to unity. In order to address the Reviewer comment, we have added the following sentence to the introduction:

“Even upon excitation of a semiconducting transducer by photons near the edge of the bandgap, heating still occurs at high fluences because of the Auger process [18].” (#8 in the list of changes)

Reviewer comment: Page 4 (middle of first paragraph) – The spot size should be given here. Also it should be mentioned that above bandgap photons are used for excitation.

The corresponding sentences are revised: “The VO₂ layer is excited by a 170 fs laser pulse focused to a spot of a 25 μm diameter with fluence W and a central photon energy of 1 eV, which is above the material bandgap of 0.6 eV [34].” (#9 in the list of changes)

Reviewer comment: Page 5 (second paragraph) – Information regarding the pump probe setup (laser type, wavelength, and rep rate) should be placed in the methods section.

We agree that information regarding the pump probe setup is expected to be found at the beginning of the Methods section. Following the reviewer’s comment, we have moved this part from the middle of the Methods section to the beginning. We have removed the information on the details of the pump probe setup from the Results section (#10-11 in the list of changes).

Reply to the Reviewer #2:

We thank the reviewer for critical reading of our manuscript and for noticing the main issue we are aiming to address in the manuscript, which is “whether phase-change materials can be practically used as an alternative of conventional metallic transducer is not well known, partly due to the limited quantitative analysis of this process. In addition, it is generally preferable to have a clean dynamical strain-induced phenomena rather than thermally induced strain. A quantitative understanding of PIPT generated strain is needed for the application of PIPT as a mechanism to generate strain pulse”. We also appreciate that the reviewer acknowledges that “this work reports interesting experiments, high-quality data, and the manuscript is well written. The methodology laid out in this work can be valuable to discuss similar process.”

Reviewer comment: The idea of strain generation via PIPT is not very new. For example, “Photoinduced Strain Release and Phase Transition Dynamics of Solid-Supported Ultrathin Vanadium Dioxide”(https://www.nature.com/articles/s41598-017-10217-0.pdf) has discussed this process. There is no significant new physics in the PIPT induced strain, which is an expected result

This is a very useful paper to understand the release of built-in strain in ultrathin VO₂ films, along with other works [40, 46-48, 49: X. He et al., *Sci. Rep.* **7**, 10045 (2017)] where the authors study the atoms’ movement by electron diffraction. These works obviously give information about generated stress and strain, but do not aim to inject any strain pulses into the substrate. The novelty of our work is to “quantify and isolate the strain contribution from multiple mechanisms” of strain pulse generation and

injection into the substrate, as pinpointed by the Reviewer. In response to the reviewer comment we have added the reference to the paper suggested by the referee to the Discussion section: "Such non-monotonic behavior has been observed earlier in some electron diffraction experiments [40, 46-48, 49: X. He et al., *Sci. Rep.* **7**, 10045 (2017), 50: A. X. Gray et al., *Phys. Rev. B* **98**, 045104 (2018)]." (#12, #30 in the List of changes).

Reviewer comment: however, it is disappointing that PIPT seems not practically generate higher strain than that in photoexcited metallic VO₂..... It is clear the involvement of PIPT does not produce larger strain than that generated in metallic VO₂. This finding significantly limits the impact of the use of PIPT for strain generation.

In response to this comment we would like to stress the following: Indeed, the strain amplitudes achieved using insulating and metallic VO₂ transducers appear to be close in value. However, as shown in Fig. 3 (c-d), the same strain amplitude is achieved with noticeably different transducer heating. It is here where the involvement of PIPT into a strain generation comes into play. Thus, at $J=3 \times 10^8$ J/m³ the higher strain is generated in the insulating transducer, while the corresponding heating is 3 times lower than in the metallic phase. The reduced heating of a transducer due to first-order PIPT, and not the absolute amplitude of the strain, is the key point in our work (see also the reply to general comment of Reviewer #1)

Reviewer comment: Secondly, I am concern about the accuracy of the estimated strain value since the strain detection via optical reflectivity is indirect (See comments 3 below). Therefore, I am afraid I won't be able to recommend this work to be published in Nature Communication due to its limited impact.

Below we address this fair concern of the Reviewer in details. We trust that resolving this and other Reviewer's concerns would convince him/her on the broad impact of our findings.

Reviewer comment: In Eq. 1 of this work, the authors seem to imply that the "thermal expansion" induced strain occurs within 1 ps, which is not accurate. In the Thomsen paper (Ref.[1]), the stress is generated via photoexcitation, (Eq. 4 in Ref. [1]). The stress can be created within 1 ps but it takes time to build the strain, which is usually limited by the sound velocity. Although this process is somewhat discussed in the Method section, it is not clear how photoexcitation leads to strain. To describe the process appropriately, it is necessary that the authors start with photoinduced stress, then discuss strain.

The reviewer outlines the correct scenario which is, in fact, described in the initial version of the manuscript. The text just before Eq.1 considers generation of stress and strain and reads: "...we consider three mechanisms which are expected to contribute to stress generation upon the optical excitation all contributions to the stress generated in VO₂ may be modeled by temporal step-like functions [1]. The net stress results in picosecond changes of the VO₂ film thickness Δa which happen during a time $\sim a/s_{VO_2}$, where $s_{VO_2}=9740$ m/s [38] is the longitudinal sound velocity in VO₂ in the direction perpendicular to the sample surface. The corresponding strain amplitude is $\epsilon_0=\Delta a/a$." In order to follow Reviewer's comment we have revised this part as follows: "The net stress results in changes of the VO₂ film thickness Δa . This change happens during a time $\sim a/s_{VO_2}$, where $s_{VO_2}=9740$ m/s [38] is the longitudinal sound velocity in VO₂ in the direction perpendicular to the sample surface. For a 100 nm thick transducer the time for the strain to emerge is then of ~ 10 ps." (#13 in the List of changes).

After that the analysis operates with strain amplitude ϵ_0 . Thus the text before Eq.1 already outlines the absolutely correct procedure required by the Reviewer #2.

Reviewer comment: The expansion of a axis as speculated in this work is not consistent with the equilibrium heating effect. The author has tried hard to argue this is plausible, but with limited experimental support. Why do the authors have to assign lattice expansion for the observed strain pulse? I would think lattice contraction can also produce strain.

The reviewer is right that contraction can also produce strain pulse. However, the shape and the fluence dependence of the measured signal clearly indicates that there is no contraction contribution to the picosecond strain pulse. We have clarified this issue in the text as follows:

In the discussion of the strain pulse analyzing upon its transmission through the sapphire (page 4 of the revised text) we specified that the simulations are performed under assumption of the VO₂ expansion: “We assume that the strain pulse injection is a result of instantaneously photo-generated stress leading to a tensile strain...” (#14 in the List of changes).

Further, we revised comparison between experimental (Fig. 2) and simulation results (page 7): “The main features of $\Delta R(t)$ are similar to those predicted in the simulations (Fig. 1d), i.e.: two negative peaks in $\Delta R(t)$ with nearly the same amplitudes are clearly seen. As the excitation density W increases, the positive part of the signal between the peaks tends to form a plateau and the signal duration τ increases. This is a reliable evidence that the laser excitation results in tensile strain in VO₂. As a result, the longitudinal strain pulse emitted into sapphire has a bipolar shape such as that shown in Fig. 1b. We note that transducer contraction would yield strain pulses with reversed polarity and the duration of the signal detected in the Cr films would decrease with W [37]. In our experiments, for both initial sample temperatures, the signal duration τ between the peaks in $\Delta R(t)$ reaches the values of $\tau=200\pm 5$ ps exceeding those measured in sapphire earlier with metallic transducers [37]. Such high values of τ point at the high amplitude $\varepsilon_0\sim 1.5$ % of the tensile strain generated in VO₂, as can be readily seen from the calibration curve $\tau(\varepsilon_0)$ in Fig. 1e. As $W\rightarrow 0$ we obtain $\tau_0\rightarrow 37$ ps at $T=295$ K, and $\tau_0\rightarrow 31$ ps at $T=355$ K. In order to reveal if the PIPT provides any substantial contribution to the generated high amplitude strain we examine in details how τ changes with increase of the excitation fluence in both insulating and metallic phases” (#15 in the List of changes).

Finally, when drawing a conclusion about the PIPT-related contribution (page 11) we revised the relevant sentence as follows: “We obtain a good agreement with the experimental results taking $\varepsilon^{PIPT} = +0.45_{-0.05}^{+0.19}\%$ possessing the same sign as ε_i and ε_e (red solid line in Fig. 3a).” (#16 in the List of changes).

We note, that this result, being inspirational for further studies by itself, does not affect the main conclusions about non-thermal contribution to the strain pulse generation. In the text we mention that non-monotonic temporal evolution of picosecond strain after photoexcitation has been observed earlier in a number of works including the Scientific Reports paper mentioned by the Reviewer.

Reviewer comment: It is not clear how accurate the estimated strain is based on optical reflectivity from Cr. The reflectivity only can infer the magnitude of the strain with a few assumptions. It is not a direct probe of absolute strain using x-ray or electron diffraction. I am not asking for time-resolved x-ray diffraction measurements but would like to know the error bar of this estimation. For example, how does the calibration curve in Fig.1e generated? Is that from simulation or experiments? Can the authors estimate the error bar of the claimed 0.45% strain arising from PIPT?

We agree with the reviewer’s statement that the accuracy of the calibration procedure is important. The most important source of uncertainty in the calibration curve arises from the fact that the sound

velocity in VO₂ in both phases remains a controversial issue. Following the request of the reviewer, we have calculated the uncertainty for Fig. 1e by using upper and lower limits for sound velocities found in literature. We also estimated the uncertainty in the strain amplitude derived from the experimental data (Fig. 3). Importantly, our initial estimates for the insulating phase are close to the lowest values still consistent with the data, indicating that the PIPT related strain might be even higher.

In order to address the Reviewer criticism, we revised Figs. 1e, 3a, 3b by adding the uncertainty limits, and their captions as follows:

Caption of Fig. 1: “Shaded areas in the panel e indicate the uncertainty ranges found by varying sound velocities for VO₂ (see Suppl. Material [Supplementary] chapter 6 for details)” (#17 in the List of changes).

Caption of Fig. 3: “Gray shaded areas in the panels a,b indicate the uncertainty ranges originating from the uncertainty in the calibration curves (Fig. 1e).” (#18 in the list of changes)

The error margins of the value $\varepsilon^{pt} = 0.45_{-0.05}^{+0.19} \%$ were calculated based on the procedure described in a new Supplementary Materials section 6 (see below). These uncertainties were added to the text in Discussion section: “We obtain a good agreement with the experimental results taking $\varepsilon^{pt} = +0.45_{-0.05}^{+0.19} \%$ possessing the same sign as ε_l and ε_e (red solid line in Fig. 3a).” (#16 in the list of changes).

We added a new chapter containing relevant discussion in the Suppl. Material (#19 in the list of changes):

”6. Error analysis of the simulation of strain pulse propagation and detection

In the model of strain pulse generation used to build the calibration curve $\tau \rightarrow \varepsilon_0$ the main uncertainty arises from elastic parameters of the VO₂ film. The parameters of the r-cut sapphire responsible for the strain pulse evolution upon propagation are known [10: J. M. Winey et al, *J. Appl. Phys.*, **90**, 3109 (2001)] with exception of viscosity. The latter is assumed to be isotropic and was taken equal to the value for c-cut Al₂O₃ [11: P. J. S. van Capel and J. I. Dijkhuis, *Phys. Rev. B*, **81**, 144106 (2010)]. In order to simulate the strain pulse injected from the VO₂ film into sapphire one needs to know mass density, light penetration depth, longitudinal sound velocity, and thickness of VO₂. Mass density and sound velocity s_{VO_2} determine reflections of the strain pulse at the interface with the sapphire substrate, while the light penetration depth and sound velocity determine initial strain pulse duration which has a pronounced impact on simulation results.

Penetration depth was calculated with optical parameters of bulk VO₂ [12: H. W. Verleur, et al., *Phys. Rev.*, **172**, 788 (1968)] which for the studied VO₂ films yielded the same absorption as measured experimentally (Fig. S5). The sound velocity was calculated using bulk VO₂ elastic constants obtained theoretically in [13: H. Dong et al., *Solid State Commun.*, **167**, 1 (2013)]. For the error estimation we considered a possible uncertainty in the latter value.

The lower limit for the sound velocity was determined by the shape of the photoelastic response (see Figs. 2a,b of the main text), namely the relative amplitude of sharp negative peaks. The decrease in the VO₂ sound velocity results in one of the peaks being smaller than the other. Taking into account experimental uncertainty and noise, we were able to estimate thus the lower limit for the VO₂ sound velocity. The upper limit for s_{VO_2} was taken to be 11500 m/s [14: E. Abreu et al, *Phys. Rev. B* **96**, 094309 (2017)] as it is the highest value ever reported in literature. Though it was reported for the metallic phase of VO₂ only, we took the same value for the dielectric phase of VO₂ as a reasonable estimation. We note here, that the higher the sound velocity, the shorter the initial strain pulse corresponding to a

particular signal duration τ , which leads to even higher estimate for the strain generated in VO₂ film for a particular absorbed energy.

This led us to the following range of sound velocities in VO₂ film on r-cut sapphire:

for insulating phase: 9300 m/s – 11500 m/s,

for metallic phase: 7500 m/s – 11500 m/s.

Using these values, we calculated the uncertainty range for the $\tau \rightarrow \epsilon_0$ calibration shown in Fig. 1e of the main text as shaded areas. This uncertainty range led to the margins of the strain value derived from the experimental data using this calibration curve (Fig. 3a,b).

The uncertainty in the calibration curve yields the margins for the $\epsilon_0(J)$ (Figs. 3a,b), i.e. this curve acquires different slopes while keeping its shape. It yields different value of deformation potential $\Xi_i = -4 \pm 0.7$ eV. The conclusion about the presence and a character of the non-thermal contribution to the strain generation remains the same, with the uncertainty of its absolute value being $\epsilon^{pt} = +0.45_{-0.05}^{+0.19}$ %.”

Reviewer comment: How about PIPT induced strain in VO₂ film growing on c-cut Al₂O₃? Does the crystal orientation matter in this process? Some discussions on this subject might be helpful.

The Reviewer makes a good point regarding the anisotropy of the shown PIPT-induced strain of 0.45 % magnitude. First, we note that the lattice parameter of VO₂ in the (100)_{M1} direction is the most affected by the phase transition, while the changes in other directions are smaller. We note this in page 4 of the initial version. Further, epitaxial growth of VO₂ film on different sapphire planes was studied in detail in a number of works. The best quality of epi-VO₂ films is achieved on the r-cut sapphire, while on the c-cut they shows larger roughness, and on m-cut – low quality epitaxy [S. Lysenko et al., *Phys. Rev. B* **75**, 075109 (2007), 36: Y. Zhao et al., *J. Appl. Phys.* **111**, 053533 (2012)]. Also, on c-cut sapphire VO₂ films exhibit 120-deg twinning, while on r-cut sapphire no twinning of the VO₂ film is observed [S. Lysenko et al., *Phys. Rev. B* **75**, 075109 (2007), 36: Y. Zhao et al., *J. Appl. Phys.* **111**, 053533 (2012), S. Lysenko et al., *J. Appl. Phys.* **114**, 153514 (2013)]. Hence, VO₂ films on the c-cut sapphire may show lower strain generation due to PIPT both due to its orientation and twinning.

We have revised the corresponding sentence in page 4 to clarify it: “The choice of VO₂ on r-cut Al₂O₃ is motivated by well-defined twin-free orientation of such films and the large change of the lattice constant along the a_{M1}(c_r) axis for the thermally-driven transition. The latter is reported to be -1 % for the bulk [35] and -0.4 % for a 120 nm film [36: Y. Zhao et al., *J. Appl. Phys.* **111**, 053533 (2012)]” (#20, #30 in the List of changes).

Further, in response to the Reviewer’s fair point, which requires further investigation, and to the question of the Reviewer #1, we added following sentence to the Conclusion: “Using transducers grown on differently oriented sapphire or other substrates may enable control over the direction in which the largest non-thermal strain generation occurs thus opening a pathway for a further optimization. This, and the complex path which the lattice takes following photo-excitation [40, 49: X. He et al., *Sci. Rep.* **7**, 10045 (2017)] suggests that VO₂ transducers with different orientations grown on various substrates may allow the generation of, not only compressive-tensile strain pulses, but also shear ones.” (#1, #30 in the List of changes).

Reply to the Reviewer #3:

We thank the reviewer for acknowledging that “this manuscript presents a proof of concept for a very nice new idea, which expands the capabilities of photo-induced strain generation” and that “the experiments are well designed and the results are well presented and analyzed.” We further appreciate detailed and constructive criticism, and that the reviewer suggests acceptance of the paper in Nature Communications after revision.

Reviewer comment: On page 10 the authors state that “the metallic phase emerges at subpicosecond time scale, which is faster than that required for the lattice “heating””, so that for $J > J_s$ the “thermoelastic contribution ϵ_{ps_0m} related to the photo-induced metallic phase only” needs to be considered. While there might be some debate about this, I think it’s a reasonable simplification. However, shouldn’t then “ $\Delta T = (J_s - \Delta J) / C_m = 28 \text{ K}$ ” be calculated using the insulating phase constants, instead of the metallic phase ones?

The discussed expression for ΔT is constructed using the following argument. ΔJ is the fraction of energy spent for PIPT, and does not contribute to thermoelastic effect, and only $J - \Delta J$ does. Since the metallic phase emerges faster than that required for the lattice “heating” [34], the $J - \Delta J$ would contribute to the thermoelastic effect in the metallic phase. Therefore, we use the constants for the metallic phase. It is explained in detail in page 10 of the main text.

Reviewer comment: “This result is different from the behavior in thermal equilibrium, when the lattice constant (in the direction of the rutile axis) is 1 % smaller in metallic than in insulating phase” (p. 12): is this also true for the c_r axis of 100 nm thin VO₂ films deposited on r-cut Al₂O₃? The reference cited is for bulk samples (and those are also the references used in section 1 of the supporting information), and strained films could behave very differently. This should be verified (either in the literature or by measuring the static lattice constant variation with T for films under study) for the comparison to the ultrafast measurements to make sense.

The Reviewer is right that in a strained film, due to the effect of the substrate, the equilibrium change of the lattice parameters may be different. In [36: Y. Zhao et al., *J. Appl. Phys.* **111**, 053533 (2012)] θ - 2θ X-ray scans are present for 120 nm VO₂ film on r-cut sapphire. These data show ~0.4% contraction upon thermal phase transition, which is smaller than in the bulk VO₂. Following the question of Reviewer #2 we note that in a few earlier works [40, 49: X. He et al., *Sci. Rep.* **7**, 10045 (2017)] nonmonotonic temporal behavior of atoms’ displacement during ultrafast PIPT is observed. We have revised the text which introduced the lattice changes upon phase transition (p. 4) as follows:

“The choice of VO₂ on r-cut Al₂O₃ is motivated by well-defined twin-free orientation of such films and the large change of the lattice constant along the $a_{M1}(c_r)$ axis for the thermally-driven transition. The latter is reported to be -1 % for the bulk [35] and -0.4 % for a 120 nm film [36: Y. Zhao et al., *J. Appl. Phys.* **111**, 053533 (2012)]”. (#20, #30 in the List of changes).

Also, the text just before the Conclusion now reads:

“This result is different from the behavior in thermal equilibrium, when the lattice constant (in the direction of the rutile axis) is smaller ($\epsilon^{pt} < 0$) in the metallic than in the insulating phase by 1 % in bulk [35] or 0.4 % in a 120 nm film [36: Y. Zhao et al., *J. Appl. Phys.* **111**, 053533 (2012)] and, correspondingly, our experiments suggest that during PIPT the lattice reaches the final compressed state non-monotonically.” (#21, #30 in the List of changes).

Reviewer comment: Assuming there is indeed a discrepancy between the sign of ϵ_{PIPT} in static vs. dynamic measurements, what could be the reason for that? It is not clearly addressed by the authors. Here it might be helpful to refer to two papers by Gray et al. (PRL 116, 116403 (2016) and PRB 98, 045104 (2018)).

The exact temporal evolution of stress and strain in VO₂ films during PIPT is a complex and nontrivial issue. Our method, being limited by the strain pulse generation processes, has access only to the first ~10 ps period of this dynamics which contributes to the strain pulse generation, rendering us unable to examine the whole process of lattice transformation in VO₂ on r-cut sapphire. In the text we refer to a number of diffraction studies that unravel non-monotonous phase transformation dynamics over longer timescale. Following the Reviewer's suggestion, we added [50: A. X. Gray et al., *Phys. Rev. B* **98**, 045104 (2018)] as a reference in this discussion: "Such non-monotonic behavior has been observed earlier in some electron diffraction experiments [40, 46-48, 49: X. He et al., *Sci. Rep.* **7**, 10045 (2017), 50: A. X. Gray et al., *Phys. Rev. B* **98**, 045104 (2018)]." (#13, #30 in the List of changes).

Minor points:

i) In Fig. 2, $t = 0$ is not the same for all the time axes. This should be clarified for clarity, as it seems like the strain pulse on the Cr film arrives before the light pulse hits the VO₂. A 28 ns time offset (in the measured traces) is mentioned only in the methods.

In response to the Reviewer comment we have revised the discussion of the data shown in Fig. 2 as follows: "Time delay denoted as $t=0$ in Fig. 2a,b corresponds to 28 ns after the moment the VO₂ film is excited by the laser pulse. This delay is equal to the time of propagation through sapphire substrate with longitudinal sound velocity. Therefore, the longitudinal strain pulse is detected". We also have added the clarifying note to the Fig. 2 caption: " $t=0$ corresponds to the time delay of 28 ns required for a strain pulse to propagate through the sapphire substrate." (#22, #23 in the list of changes).

ii) Page 10: "It is natural to approximate the increase of the fraction" => worth providing a bit more detail on why this is "natural"? Perhaps relating it to the Avrami model (with exponent 2), which is very similar to erf?

We thank the Reviewer for mentioning the Avrami model relevant for nucleation and growth of a new phase which, for phase transition with zero nucleation rate in thin VO₂ film (2D in-plane growth), would yield Avrami exponent 2. This model indeed provides an expression for a fraction of transformed material, but shows evolution in a time domain whereas for Fig. 3 a relation of the fraction of the nucleated new phase to the absorbed energy J is required. We acknowledge that Kolmogorov-Johnson-Mehl-Avrami model has a lot of modifications and applications for a thermally-induced phase transitions [J. Farjas, P. Roura, *Acta Materialia* **54**, 5573–5579 (2006)] but to the best of our knowledge it is a challenge to modify this model for a photo-induced ultrafast phase transition and complex kinetics involved. So our reasoning was not based on PIPT kinetics but rather on distribution of nucleation sites in the sample. This distribution was assumed to be Gaussian as a reasonable approximation for an unknown probabilistic quantity, which, upon integration, yields the erf-function in the expression (eq. 3). To address the criticism of the Reviewer we included aforementioned explanation to the text on page 11, which now reads: "The increase of the fraction of the excited material undergoing the PIPT from 0 to 1 as the absorbed energy increases from J_1 to J_5 is related to a distribution of nucleation sites

in the sample versus energy which was approximated by the Gaussian error function centered at $J_0=(J_T+J_S)/2$, with a dispersion parameter σ_0 ." (#24 in the list of changes).

iii) "At an excitation close to the saturation of PIPT, $J \sim J_S$, $\epsilon_0(J)$ flattens off" (p. 7); "temperature rise stops between J_T and J_S " (p. 11): none of these is very precise, consider rephrasing.

We have revised the corresponding sentences as follows: "As an excitation exceeds the saturation of PIPT, $J > J_S$, $\epsilon_0(J)$ resumes linear dependence with a slope steeper than that for $J < J_T$ " (#25 in the List of changes).

"When the VO_2 is excited being initially in its insulating phase (Fig. 3c), the temperature rise dT/dJ decreases significantly between J_T and J_S , while the generated strain still increases due to non-thermal PIPT-related ϵ^{pt} and the electronic $\epsilon_0^e(J)$ contributions." (#26 in the list of changes).

iv) "the film expands during PIPT, at least over the first 10 ps" (p. 12): where does the 10 ps timescale come from?

The estimate is obtained based on the sound velocity in VO_2 $s_{VO_2} \sim 9500$ m/s and VO_2 film thickness $a=100$ nm. We agree with the Reviewer that this part in the text needs clarification and rewrite it as follows: "the film expands during PIPT, at least over the time of the strain pulse generation $a/s_{VO_2} \sim 10$ ps" (#27 in the list of changes).

v) "At first, after PIPT, the VO_2 film expands along the rutile axis during a short time $\sim a/s$ and then shrinks more slowly until thermodynamic equilibrium is reached." (p. 12): the shrinking part of the dynamics is not discussed before, how can it be seen from the data?

The slower shrinking part of the lattice dynamics does not contribute to the picosecond strain pulse and thus cannot be detected in our experiments. The strain pulse detected in our experiments unambiguously points to the expansion of the film during the first several ps, and no fingerprint of shrinking is seen, indicating that the latter proceeds on a longer time scale and cannot lead to the strain pulse generation.

REVIEWERS' COMMENTS:

Reviewer #1 (Remarks to the Author):

The authors addressed all of my concerns in the revised manuscript.

Reviewer #2 (Remarks to the Author):

In the response letter and revised manuscript, the authors addressed my technical concerns well. For example, the estimated strain error is very helpful. In addition, I found the estimated strain as they reported agrees well with direct XRD measurements shown in Fig. 4 of PRB, 88, 165424 (2013) (please cite).

The only issue is still the novelty. The work does not really contain significantly new physics. The concept of using VO₂ as a strain transducer is not novel either. The quantification of multiple sources of strain arising during photoinduced phase transition (PIPT) is very useful but is an incremental extension of the concept. However, I agree that this is a nice, may also be the first systematic study on strain launched from phase change materials into the substrate. The demonstrated technique to characterize transient strain is quite new up to my knowledge. This strain characterization method to me is probably even more valuable than PIPT induced strain process discussed in their work. Balancing the above points, I am not totally certain if this work warrants the publication on Nature Communication, but I am not against its publication either.

We thank the Referees for considering the revised manuscript. We are happy that the Referees found our responses and the revision of the manuscript satisfactory. We appreciate that the Reviewer #2 acknowledges the novelty of our approach and does not object the acceptance of the manuscript.

Below we address the final issue raised by the Reviewer #2.

Reviewer comment: 'In addition, I found the estimated strain as they reported agrees well with direct XRD measurements shown in Fig. 4 of PRB, 88, 165424 (2013) (please cite).'

We are grateful for the Reviewer for mentioning this paper which supports results obtained in our work. We added a sentence in *Discussion* Section on page 13 to include this citation:

'The fast initial expansion of VO₂ thin film along a_{M1} axis up to 0.4% due to PIPT was also observed previously by ultrafast x-ray diffraction [46].'

We also added new reference advised by the Reviewer:

46. H. Wen, Lu Guo, E. Barnes, J. H. Lee, D. A. Walko, R. D. Schaller, J. A. Moyer, R. Misra, Y. Li, E. M. Dufresne, D. G. Schlom, V. Gopalan, and J. W. Freeland. Structural and electronic recovery pathways of a photoexcited ultrathin VO₂ film. *Phys. Rev. B*, **88**, 165424 (2013).

On behalf of co-authors

Sincerely yours

Iaroslav Mogunov